# *Castanopsis sieboldii* Extract Alleviates Acute Liver Injury by Antagonizing Inflammasome-Mediated Pyroptosis

**DOI:** 10.3390/ijms241511982

**Published:** 2023-07-26

**Authors:** Jae Min Kim, Sam Seok Cho, Sohi Kang, Changjong Moon, Ji Hye Yang, Sung Hwan Ki

**Affiliations:** 1College of Pharmacy, Chosun University, Gwangju 61452, Republic of Korea; kjm4994@naver.com (J.M.K.); messicho@naver.com (S.S.C.); 2College of Veterinary Medicine, Chonnam National University, Gwangju 61186, Republic of Korea; shrloveu@chonnam.ac.kr (S.K.); moonc@chonnam.ac.kr (C.M.); 3College of Korean Medicine, Dongshin University, Naju 58245, Republic of Korea

**Keywords:** *Castanopsis sieboldii*, hepatitis, Kupffer cells, LPS, inflammasome, pyroptosis

## Abstract

*Castanopsis sieboldii* (CS), a subtropical species, was reported to have antioxidant and antibacterial effects. However, the anti-inflammatory effects of CS have not been studied. This study aimed to investigate whether the 70% ethanol extract of the CS leaf (CSL3) inhibited lipopolysaccharide (LPS)-induced inflammatory responses and LPS and ATP-induced pyroptosis in macrophages. CSL3 treatment inhibited NO release and iNOS expression in LPS-stimulated cells. CSL3 antagonized NF-κB and AP-1 activation, which was due to MAPK (p38, ERK, and JNK) inhibition. CSL3 successfully decreased NLRP3 inflammasome activation and increased IL-1β expression. CSL3 treatment diminished LPS and ATP-induced pore formation in GSDMD. The in vivo effect of CSL3 on acute liver injury was evaluated in a CCl_4_-treated mouse model. CCl_4_ treatment increased the activity of serum alanine aminotransferase and aspartate aminotransferase, which decreased by CSL3. In addition, CCl_4_-induced an increase in TNF-α, and IL-6 levels decreased by CSL3 treatment. Furthermore, we verified that the CCl_4_-induced inflammasome and pyroptosis-related gene expression in liver tissue and release of IL-1β into serum were suppressed by CSL3 treatment. Our results suggest that CSL3 protects against acute liver injury by inhibiting inflammasome formation and pyroptosis.

## 1. Introduction

The liver is the largest solid organ in the human body and is responsible for xenobiotic metabolism and detoxification. The liver mass is composed of 80% parenchymal cells (hepatocytes) and 20% non-parenchymal cells (Kupffer cells, lymphocytes, hepatic stellate cells, and endothelial cells) [1]. Kupffer cells, liver-resident macrophages, play a central role in the pathogenesis of a variety of liver diseases, including alcoholic liver disease, nonalcoholic steatohepatitis, intrahepatic cholestasis, viral hepatitis, and liver fibrosis [2]. During liver injury, activated Kupffer cells are a major source of inflammatory mediators, including nitric oxide, cytokines, chemokines, superoxide, and eicosanoids [3], resulting in excessive inflammatory response and hepatitis.

Hepatitis can be classified as acute or chronic based on the duration of inflammation in the liver. The majority of patients with acute hepatitis recover spontaneously within several days [4]; however, treatment is necessary if symptoms persist. The treatment of choice for patients with severe alcoholic hepatitis (AH) or autoimmune hepatitis is corticosteroids. However, the use of corticosteroids in patients with acute hepatitis is still controversial [5]. In contrast, chronic hepatitis can cause chronic liver diseases, including liver fibrosis, cirrhosis, and liver cancer, leading to significant morbidity and mortality [6]. Current treatments for late-stage liver diseases, such as decompensated cirrhosis and hepatocellular carcinoma, are still limited, and liver transplantation is the only available approach to improve survival. Thus, the identification of new drug targets and candidate drugs that can regulate the early stages of liver diseases, such as hepatitis, is crucial.

Kupffer cells are responsible for immune and repair functions in the liver and are polarized into the M1 (a pro-inflammatory form) or M2 phenotype (an anti-inflammatory form) [7]. Polarization into the M1 phenotype induced by IFN-γ, TNF-α, and/or lipopolysaccharide (LPS) increases pro-inflammatory cytokine release, which is associated with liver inflammation. When Kupffer cells are stimulated by LPS and/or IFN-γ, the NLR family pyrin domain containing 3 (NLRP3) induces the inflammasome formation and cleavage of gasdermin D (GSDMDC1), leading to pyroptosis. Pyroptosis is a type of lytic programmed cell death that is stimulated by proinflammatory signals [8,9]. Accumulating evidence shows that pyroptosis in Kupffer cells also contributes to sterile inflammatory liver diseases by increasing the release of pro-inflammatory cytokines, including IL-18 or IL-1β [10]. However, there are no candidate drugs for the prevention or treatment of acute hepatitis that target pyroptosis in Kupffer cells. 

*Castanopsis sieboldii* (CS), a warm-temperature tree species, is native to the southern part of the Korean Peninsula and Jeju Island. Warm-temperature tree species distributed in subtropical regions are gradually expanding on the Korean Peninsula due to global warming and climate change. The fruits of CS have been used as food since ancient times, and their branches have been recently used as wood. In addition, the leaf extract of CS has high antioxidant and antibacterial activities [11]. Recently, we determined that the 70% ethanol (EtOH) extract of the CS leaf (CSL3) has cytoprotective and antioxidant effects in keratinocytes against UV-irradiated photodamage [12]. However, the effects of CSL3 on acute hepatitis and Kupffer cell activation remain unknown. 

Therefore, the current study investigated whether CSL3 inhibits inflammatory responses and inflammasome-mediated pyroptosis in activated Kupffer cells. Furthermore, we examined the in vivo effects of CSL3 on CCl_4_-treated acute hepatitis mice by measuring the activities of marker enzymes for hepatic functioning in plasma and analyzing histopathological profiles of hepatic damage. 

## 2. Results

### 2.1. Suppression of LPS-Induced Inflammatory Response by CSL3 

Kupffer cells, also known as Kupffer–Browicz cells, are liver-resident macrophages located in the lumen of the liver sinusoids. It was well-established that Kupffer cells (KCs), resident liver macrophages, play a major role in the pathophysiology of acute liver injury [13,14]. Thus, ImKC cells, a murine immortalized Kupffer cell line, were adopted to investigate the effect of CSL3 on acute liver injury and its molecular mechanism. We also used the RAW264.7 cell line, the most popular murine macrophage cell line, to confirm the effect of CSL3 on ImKC cells. First, we evaluated the cytotoxicity of CSL3 in both cell lines using a WST-1 cell viability assay. There was no cytotoxicity at CSL3 concentrations of up to 100 μg/mL in either cell line (Figure 1A,B). 

Induction of iNOS is a direct outcome of the inflammatory response [15]. Therefore, we investigated whether CSL3 can inhibit LPS-induced iNOS expression in ImKCs and RAW264.7 cells. The LPS-stimulated iNOS protein levels were decreased by CSL3 in a dose-dependent manner (Figure 1C,D). In agreement with protein expression, CSL3 inhibited LPS-induced mRNA levels of *iNOS* in ImKCs and RAW264.7 cells (Figure 1E,F). Also, CSL3 treatment suppressed the increase in iNOS promoter activity by LPS in stably transfected RAW264.7 cells (Figure 1G). NO produced by iNOS is involved in DNA damage and aberrant cell signaling in various tissues [16]. The LPS-induced increase in NO production was reduced by CSL3 treatment in a dose-dependent manner in ImKCs (Figure 1H). The effects of CSL3 were comparable in both cells; further experiments were thus conducted only in ImKC cells.

Next, we investigated whether CSL3 inhibits the expression of inflammatory cytokines and genes. The RT-PCR analysis showed that the increased inflammatory gene expression of *Il-6* and *TNF-α* was inhibited by CSL3 (Figure 2A). The release of cytokines by LPS into the medium was abolished by CSL3. The elevated cytokine levels were also significantly reduced by CSL3 treatment (Figure 2B,C). Our results showed that CSL3 alleviated LPS-induced inflammatory gene expression in the macrophages. 

### 2.2. Inhibition of LPS-Induced Inflammatory Signaling Pathway by CSL3 

NF-κB and AP-1 can be activated by a variety of inflammatory stimuli and regulate the transcriptional activation of inflammatory genes [17,18]. Nuclear translocation of NF-κB is activated by phosphorylation and subsequent degradation of IκB-α [19]. Phosphorylation and degradation of IκB-α were increased by LPS but were completely blocked by pretreatment with CSL3 (Figure 3A,B). Furthermore, treatment with CSL3 inhibited the LPS-induced increase in nuclear translocation of p65 protein levels (Figure 3C). The inducible transcriptional complex AP-1 (composed of c-Jun and c-Fos proteins) is also important for cellular adaptation in response to many exogenous stimuli [20]. To determine the inhibitory effect of CSL3 on LPS-induced AP-1 activation, a western blot analysis was performed. LPS increased the phosphorylation of c-Jun and c-Fos in the cell lysates, which were markedly inhibited by CSL3 pretreatment in ImKC cells (Figure 3D). 

MAPKs, including ERK, p38, and JNK, initiate a downstream signaling cascade of transcription factors, including NF-κB and AP-1, for pro-inflammatory gene expression [21]. Accordingly, we investigated whether NF-κB and AP-1 inhibition by CSL3 was due to the MAPK signaling pathway. The phosphorylation of MAPKs (JNK, ERK, and p38) was increased by LPS, whereas CSL3 decreased JNK, ERK, and p38 phosphorylation in the cells (Figure 3E). These results indicate that the anti-inflammatory effect of CSL3 may be associated with the suppression of NFκB and AP-1 via MAPK signaling. 

### 2.3. Attenuation of Inflammasome-Mediated Pyroptosis by CSL3 

Pyroptosis, a programmed cell death caused by inflammation, has some similarities with apoptosis, including the occurrence of DNA damage and chromatin condensation [22]. However, pyroptosis possesses some unique characteristics that are distinct from apoptosis [23]. Pyroptosis is mediated by inflammasome accumulation, which is accompanied by NLRP3 activation, the cleavage of caspase-1 and GSDMD, and IL-1β release [24]. Therefore, we investigated whether CSL3 inhibited NLRP3 inflammasome activation and pyroptosis induced by LPS and ATP in ImKCs. LPS and ATP treatments increased IL-1β maturation, the cleavage of caspase-1 and GSDMD, and NLRP3 expression, whereas CSL3 treatment decreased inflammasome formation (Figure 4A). In addition, CSL3 suppressed LPS and ATP-induced IL-1β mRNA expression and IL-1β release into the medium in ImKC cells (Figure 4B,C). These results suggest that the cytoprotective effect of CSL3 is due to the inhibition of inflammasome-mediated pyroptosis.

### 2.4. Protective Effect of CSL3 on CCl_4_-Induced Acute Hepatitis in Mice

CCl_4_-induced acute hepatitis is the most popular in vivo model for studying intrinsic chemical-induced liver injury [25,26]. Therefore, we adopted a CCl_4_-induced acute liver injury model to investigate the hepatoprotective effects of CSL3. The mice were pretreated with CSL3 five times before a CCl_4_ injection (Figure 5A, upper panel). The CCl_4_ treatment group showed hepatic injury characteristics. However, the CSL3 group showed less hepatic damage (Figure 5A, lower panel). Serum ALT and AST activities were increased by CCl_4_ treatment but were significantly reduced by CSL3 treatment (Figure 5B). Similarly, H&E staining of the liver tissue showed that liver injury caused by CCl_4_ was chiefly caused by lesions around the central vein, but CSL3 treatment protected against histological damage (Figure 5C). 

Next, the levels of inflammatory cytokines (TNF-α and IL-6) in the mouse serum were investigated using an ELISA kit. The levels of TNF-α and IL-6 were significantly diminished by CSL3 treatment (Figure 6A,B). Moreover, CSL3 treatment markedly decreased iNOS protein expression (Figure 6C). 

### 2.5. Inhibition of Inflammasome-Mediated Pyroptosis by CSL3 in CCl_4_-Treated Mice

It was recently reported that CCl_4_ leads to acute liver injury via suppression of the antioxidant pathway and NLRP3 inflammasome activation [27,28]. Therefore, we verified whether CCl_4_-induced inflammasome and pyroptosis marker protein expression (NLRP3 and cleaved caspase-1 and GSDMD) were suppressed by CSL3 treatment (Figure 7A). CSL3 treatment inhibited inflammasome-mediated pyroptosis by CCl_4._ CCl_4_-induced IL-1β release in serum was significantly attenuated by CSL3 treatment (Figure 7B). These results suggest that the hepatoprotective effect of CSL3 is due to the attenuation of inflammasome-mediated pyroptosis in Kupffer cells.

## 3. Discussion

CS fruits have been used as food and its branches as wood. In addition, it was reported that the leaf extract of CS (CSL) had high antioxidant and antibacterial activities. Previously, we succeeded in extracting CS under several conditions and found that the 70% EtOH extract of CSL (CSL3) had the highest antioxidant activity [12]. Moreover, we found that CSL3 protected against reactive oxygen species (ROS) production and cell death in UVB-irradiated keratinocytes by inhibiting ER stress and activating autophagy. However, the pharmacological efficacy of CSL3 in acute hepatitis remains unclear. Therefore, we verified that CSL3 exerts a protective effect by suppressing inflammatory responses and inflammasomes in in vitro and in vivo models of acute hepatitis. 

We first examined whether CSL3 has anti-inflammatory effects on LPS-activated macrophages (ImKC: immortalized murine Kupffer cells and Raw264.7 cell: macrophage cell line from a tumor male mouse induced with the Abelson leukemia virus). CSL3 significantly inhibited inflammatory responses characterized by NO production and iNOS expression in both macrophages (Figure 1). The anti-inflammatory effect of CSL3 was comparable in both macrophages, and further experiments were carried out only in ImKC cells. CLS3 treatment antagonized the LPS-induced increase in inflammatory gene expression and pro-inflammatory cytokine levels (e.g., IL-6 and TNF-α) (Figure 2) in ImKC cells. LPS binds and activates toll-like receptor 4 (TLR4, CD284), which causes the phosphorylation and degradation of IκB-α [29]. Degradation of IκBα leads to NF-κB activation, causing the translocation of p65 (a subunit of NF-κB transcription complex) into the nucleus to produce NO and pro-inflammatory cytokines [17]. We showed that CSL3 mitigated NF-κB and AP-1 activation, which was mediated by the inhibition of MAPK activation (ERK, JNK, and p38) (Figure 3). These results suggested that CSL3 inhibits Kupffer cell activation and inflammatory responses via blocking MAPK/NF-κB/AP-1.

We then examined whether CSL3 could inhibit the LPS and ATP-induced inflammasome and pyroptosis in ImKC. Macrophage activation by LPS can cause an increase in oxygen uptake, resulting in the production of a range of reactive oxygen species, which are the key factors driving oxidative stress-triggered inflammation in immune and inflammatory cells [30]. In the extracellular environment, ATP is markedly released in response to tissue damage and cellular stress [31]. ATP, which is generally secreted from dying and stressed cells, is used as a damage-activated molecular pattern (DAMP) to activate NLRP3 inflammasome formation [32,33]. Canonical pyroptotic cell death is mediated by NLRP3 inflammasome accumulation, which is accompanied by GSDMD cleavage and IL-1β and IL-18 release [34]. The NLRP3 inflammasome triggers caspase-1 activation, which results in the cleavage of GSDMD, allowing the N-terminal domain to oligomerize and form pores in the cell membrane, causing cell membrane rupture [35]. Treatment with LPS and ATP induces inflammasome accumulation and pyroptosis in Kupffer cells. However, CSL3 pretreatment inhibited the expression of NLRP3, an inflammasome sensor, and blocked the cleavage of caspase-1 and GSDMD. CSL3 also decreased the production of pro-inflammatory cytokines, such as IL-1β, related to inflammasomes and pyroptosis (Figure 8). There are some reports of natural compounds with mechanisms similar to CSL3, which prevent NLRP3 inflammasome [36,37,38]. Nevertheless, few studies focused their effects on acute liver injury due to inhibiting Kupffer cell inflammasome. 

Finally, we investigated whether CSL3 protects against acute liver injury in mice with CCl_4_-induced hepatitis. CCl_4_ is widely used in animal models to induce acute hepatitis and liver fibrosis [39,40]. The increased levels of serum ALT and AST induced by CCl_4_ administration were diminished by CSL3, and H&E staining results showed that liver damage was attenuated by CSL3 (Figure 5). Moreover, CSL3 treatment inhibited the production of inflammatory cytokines (IL-6 and TNF-α) in the serum and expression of iNOS in the liver tissue (Figure 6). Recently, CCl_4_ was reported to induce hepatitis via inflammasome formation [25,28]. We then examined whether CSL3 inhibited the expression of inflammasome and pyroptosis-related proteins, such as NLRP3, GSDMD, and caspase-1, and significantly reduced the production of IL-1β in serum (Figure 7). These results indicate that CSL3 treatment restored CCl_4_-induced liver inflammation and injury by inhibiting inflammasome-mediated pyroptosis.

We found that CSL3 contained the highest content of phenolic compounds, and the major constituent of CSL3 was epigallocatechin-3-gallate (EGCG), a well-known antioxidant and anti-inflammatory polyphenol, by HPLC and LC-MS/MS analysis [12]. Surprisingly, the EGCG content in CS was much higher than in green tea. It has been reported that EGCG prevents liver inflammation, oxidative stress, and even fibrosis in CCl_4_-induced liver injury in mice [41]. EGCG treatment suppressed NLRP3 inflammasome in Kupffer cells, which resulted in the therapeutic effect of EGCG against HBV-induced liver injury [42]. Therefore, we speculated that the anti-inflammatory and hepatoprotective effects of CSL3 are due to EGCG. 

Although we could observe the hepatoprotective effect of CSL3 in Kupffer cells and CCl_4_-injected mice, there are some limitations in the current study. (1) We previously reported that CSL3 had the highest antioxidative capacity. We thus cannot exclude the possibility of whether the antioxidative efficacy of CSL3 might contribute to preventing acute hepatitis. (2) Although CSL3 could succeed in inhibiting the inflammasome signaling pathway, we do not have any direct clues whether the CSL3 effect was due to inflammasome inhibition. Biochemical and genetic experimental models were still required to unveil the concise molecular mechanism of CSL3. (3) It was reported that EGCG, a major constituent of CSL3, inhibited the inflammasome signaling pathway. However, further studies are necessary to define whether EGCG and other components might contribute to the CSL3 effect against acute hepatitis. These are currently being conducted by this research team as a separate project.

In conclusion, our data indicate that CSL3 exerts hepatoprotective effects against CCl_4_-induced acute liver injury via the suppression of inflammation and pyroptosis. Based on these results, we propose that CSL3 might be a potential therapeutic drug candidate for the prevention and treatment of acute hepatitis.

## 4. Materials and Methods

### 4.1. Materials

IκB-α, iNOS, β-tubulin, p65, and NLRP3 antibodies were acquired from Santa Cruz Biotechnology (Dallas, TX, USA). Antibodies against phospho-JNK1/2, JNK1/2, phospho-ERK1/2, ERK1/2, phospho-p38, p38, phospho-IκB-α, lamin A/C, and cleaved-caspase-1 were obtained from Cell Signaling Technology (Danvers, MA, USA). The IL-1β antibody was provided by R&D Systems (Minneapolis, MN, USA). The gasdermin D antibody was purchased from NOVUS Biologicals (Littleton, CO, USA). ATP, LPS (*Escherichia coli* 055:B5). The β-actin antibody and sodium nitrite were acquired from Sigma Chemicals (St. Louis, MO, USA). The horseradish peroxidase-conjugated goat anti-rabbit, anti-goat, and anti-mouse antibodies were obtained from Invitrogen (Carlsbad, CA, USA).

### 4.2. Preparation of 70% Ethanol Extracts of CSL3

CSL3 used in this study was collected around April 2020. It was supplied by Wando Arboretum (Wando-gun, Korea), washed, dried with hot air, stored, and ground. Dried and ground CSL3 (1.5 kg) was added to 70% EtOH (*v/v*) (15.0 L) and immersed at room temperature for 2 weeks for extraction. The immersed sample was filtered using a vacuum filtration device and Whatman No. 1, and the residue separated by this method was collected once under the same conditions. The filtrate obtained was concentrated in a rotary vacuum evaporator at 37–40 °C, freeze-dried, and used.

### 4.3. Cell Culture 

ImKCs (SCC119) were obtained from Sigma-Aldrich, and RAW 264.7 cells were purchased from American Type Culture Collection (ATCC; Rockville, MD, USA). The cells were grown in Dulbecco’s modified eagle medium (DMEM) (high glucose) supplemented with 50 units/mL penicillin/streptomycin and 10% fetal bovine serum at 37 °C in a humidified 5% CO_2_ atmosphere. The cells were serum-starved for 12 h before treatment, and the control group was treated with vehicle correspondence. To induce inflammation in macrophages, the cells were treated with 0.1 μg/mL LPS, as previously reported [43,44,45]. Moreover, LPS (0.1 μg/mL)/ATP (5 mM) was adopted to induce inflammasome in macrophages, as previously reported [46].

### 4.4. WST-1 Cell Viability Assay

Cell viability was estimated using an EZ-Cytox kit (DoGenBio Co., Ltd., Seoul, Korea). The cells were plated in 12-well plates, incubated with CSL3 alone for 12 h, and then the medium was changed with a 5% WST-1 reagent for 0.5 h. Then, 100 μL of the cell medium was transferred to a microplate. Absorbance at 450 nm (600 nm was used as the reference wavelength) was measured using a microplate reader (Molecular Device, San Jose, CA, USA).

### 4.5. NO Production Assay

NO production was evaluated using a Griess reaction (Sigma, St. Louis, MO, USA). Following the pretreatment of cells with CSL3 for 1 h, the cells were incubated with LPS for 15 h. The cell medium (100 μL) was transferred into a microplate, and then a 100 μL Griess reagent (0.04 g/mL dissolved in distilled water (DW)) was added and allowed to react at room temperature for 0.5 h. Absorbance at 540 nm was analyzed using a microplate reader.

### 4.6. Immunoblot Analysis

Subcellular fractionation, protein extraction, sodium dodecyl sulfate-polyacrylamide gel electrophoresis (SDS-PAGE), and an immunoblot analysis were performed as previously described [47]. Protein extracts were separated into 7.5% and 12% gels by electrophoresis and then transferred to nitrocellulose membranes. After membrane blocking was performed using 5% skimmed milk for 0.5 h, they were incubated overnight with primary antibodies at 4 °C and 20 rpm and then incubated with a horseradish peroxidase-conjugated secondary antibody. The immunoreactive proteins were visualized using an enhanced chemiluminescence (ECL) detection kit. β-actin was used as a control, and β-tubulin or lamin A/C was used to verify the integrity of subcellular fractionation.

### 4.7. RNA Isolation and RT-PCR Analysis

Total RNA was isolated using a TRIzol reagent (Invitrogen), according to the manufacturer’s instructions. cDNA was prepared using a cDNA synthesis kit (Bioneer, Daejeon, Korea) and a thermal cycler (Bio-Rad, Hercules, CA, USA). To obtain cDNA, total RNA (2 μg) was reverse transcribed using an oligo (dT)_18_ primer. Primers were synthesized by Bioneer. The following primer sequences were used: mouse iNOS sense, 5′-CCTCCTCCACCCTACCAAGT-3′ and antisense, and 5′-CACCCAAAGTGCTTCAGTCA-3′; mouse TNF-α sense, 5′-AAGCCTGTAGCCCACGTCGTA-3′ and antisense, and 5′-AGGTACAACCCATCGGCTGG-3′; mouse IL-6 sense, 5′-TCCATCCAGTTGCCTTCTTG-3′ and antisense, and 5′-TTCCACGATTTCCCAGAGAAC-3′; mouse IL-1β sense, 5′-TGGACGGACCCCAAAAGATG-3′ and antisense, and 5′-AGAAGGTGCTCATGTCCTCA-3′; and mouse GAPDH sense, 5′-TGCCCCCATGTTTGTGATG-3′ and antisense, and 5′-TGTGGTCATGAGCCCTTCC-3′. GAPDH was used as a control for RT-PCR.

### 4.8. Reporter Gene Assay

To measure the activities of the *iNOS* promoter constructs, luciferase reporter assays were performed in cells stably transfected with the *iNOS* gene promoter pGL-miNOS-1588, which contains the murine *iNOS* promoter from –1588 to +165 bp, as previously described [48]. For the luciferase assay, the cells were incubated with 0.1 μg/mL LPS in the presence or absence of CSL3 for 6 h. After discarding the medium, a passive lysis buffer (Promega, Madison, WI, USA) was immediately added to the cells. Luciferase assay reagent II (Promega, Madison, WI, USA) was added to the lysates, and luciferase activity was analyzed using a luminometer (Promega, Madison, WI, USA). The relative *iNOS* promoter-driven luciferase activity was calculated by normalizing luciferase activity to the protein concentration.

### 4.9. Enzyme-Linked Immunosorbent Assay (ELISA)

IL-6, IL-1β, and TNF-α levels were quantified using an ELISA kit (Invitrogen, Waltham, MA, USA), according to the manufacturer’s instructions. IL-6, IL-1β, and TNF-α levels in the cell supernatant or serum were analyzed by ELISA using anti-mouse IL-1β, TNF-α, or IL-6 antibodies and biotinylated secondary antibodies, according to the manufacturer’s instructions.

### 4.10. Animals and Treatment 

The protocols of all animal experiments were reviewed and approved by the Animal Care and Use Committee of Dongshin University (DSU2021-01-06). We purchased 6-week-old ICR male mice from Oriental Bio (Sungnam, Korea), and the mice were allowed to adapt to the lab conditions for 1 week. The mice (5 mice per group) were housed at 20 ± 2 °C under pathogen-free air filtered at a 12 h light/dark cycle and relative humidity of 50 ± 5%. The mice were supplied with chow (G-bio, Gwangju, Korea) and water. The mice were orally administered CSL3 (250 or 500 mg/kg CSL3) dissolved in 40% polyethylene glycol (PEG) for 5 days. To induce acute hepatitis, CCl_4_ (1 mL/kg) dissolved in olive oil (10%) was intraperitoneally injected into the mice, as previously reported [49,50]. Food intake and body weight were monitored daily and were not different between the groups. The mice were sacrificed after 24 h, and blood and liver samples were collected. 

### 4.11. Blood Chemistry 

Serum alanine aminotransferase (ALT) and aspartate aminotransferase (AST) levels were analyzed using commercial kits (Asan Pharmaceutical, Seoul, Korea).

### 4.12. Histopathological Examination 

The mouse liver tissue samples were fixed in a 10% formalin buffer, embedded in paraffin wax, and cut into 3 μm thick sections, which were stained with hematoxylin and eosin (H&E) for routine examination. To estimate histopathological changes, stained tissue samples were identified under a light microscope. An arbitrary scope was applied to each microscopic field viewed at magnifications of 40–200×. 

### 4.13. Statistical Analysis

One-way analysis of variance (ANOVA) was used to evaluate the statistical significance of the differences among the treatment groups. For each statistically significant treatment effect, the Newman–Keuls test was used for comparisons between multiple group means. The data are expressed as mean ± standard error (S.E.).

## Figures and Tables

**Figure 1 ijms-24-11982-f001:**
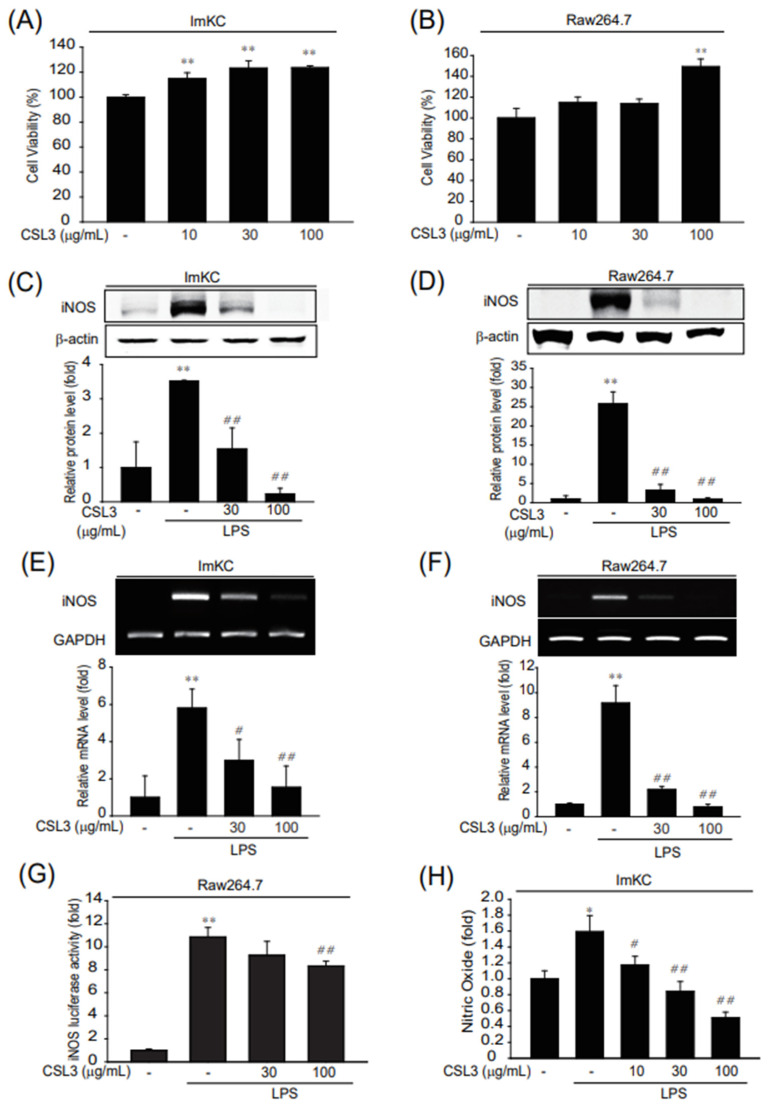
The enhibitory effect of CSL3 on iNOS expression and NO production in LPS-activated Kupffer cells. (**A**,**B**) The effect of CSL3 (10–100 μg/mL, 24 h) on cytotoxicity was estimated using WST-1 assays in ImKCs and Raw264.7 cells. Data represent the mean ± S.E. of three replicates; ** *p* < 0.01, significant versus vehicle-treated control. (**C**,**D**) The expression of iNOS protein in macrophages. ImKC (**C**) or Raw264.7 cells (**D**) were pretreated with CSL3 for 0.5 h, treated with LPS, and incubated for 12 h. iNOS protein expression was visualized by immunoblotting and results were confirmed by repeated experiments. (**E**,**F**) The enhibition of *iNOS* mRNA by CSL3 treatment in macrophages. ImKC (**E**) or Raw264.7 cells (**F**) were pretreated with CSL3 for 1 h and treated with LPS. After 3 h, the expression level of iNOS was evaluated by PCR analysis, and the results were confirmed by repeated experiments. (**G**) iNOS luciferase assays were performed in cells stably transfected with pGL-miNOS-1588, which contains murine iNOS promoters from −1588 to +165bp and exposed to LPS and/or CSL3 in cells for 12 h. (**H**) NO production was measured by a Griess reagent. ImKCs were pretreated with CSL3 for 1 h and then incubated with LPS for 15 h. Data represent the mean ± S.E. of three replicates; * *p* < 0.05, ** *p* < 0.01, significant versus vehicle-treated control; ^##^
*p* < 0.01, ^#^
*p* < 0.05, significant versus LPS alone.

**Figure 2 ijms-24-11982-f002:**
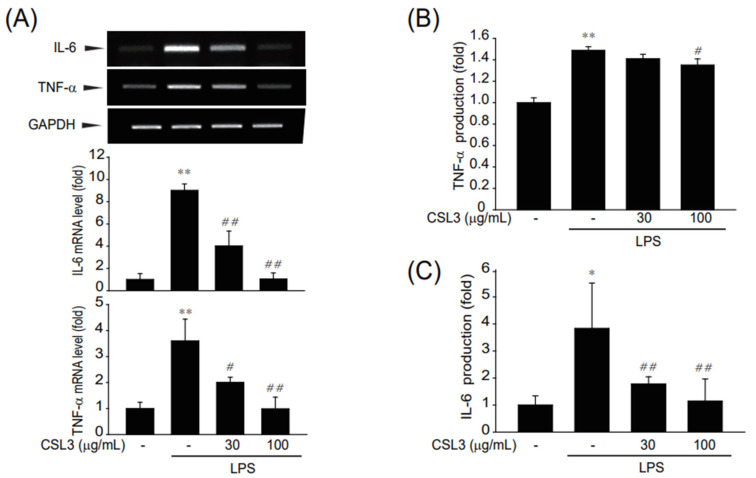
Suppression of inflammatory gene expression by CSL3 in LPS-activated Kupffer cells. ImKC cells were pretreated with CSL3 for 1 h and then treated with LPS for 3 h. (**A**) mRNA levels of inflammatory cytokines (*Il-6* and *TNF-α*) were evaluated by RT-PCR using GAPDH as a control, and the results were verified by three repeated experiments. (**B**,**C**) IL-6 and TNF-α release into the cell supernatant was determined by ELISA. Data represent the mean ± S.E. of three replicates; ** *p* < 0.01, * *p* < 0.05, significant versus vehicle-treated control; ^##^
*p* < 0.01, ^#^
*p* < 0.05, significant versus LPS alone.

**Figure 3 ijms-24-11982-f003:**
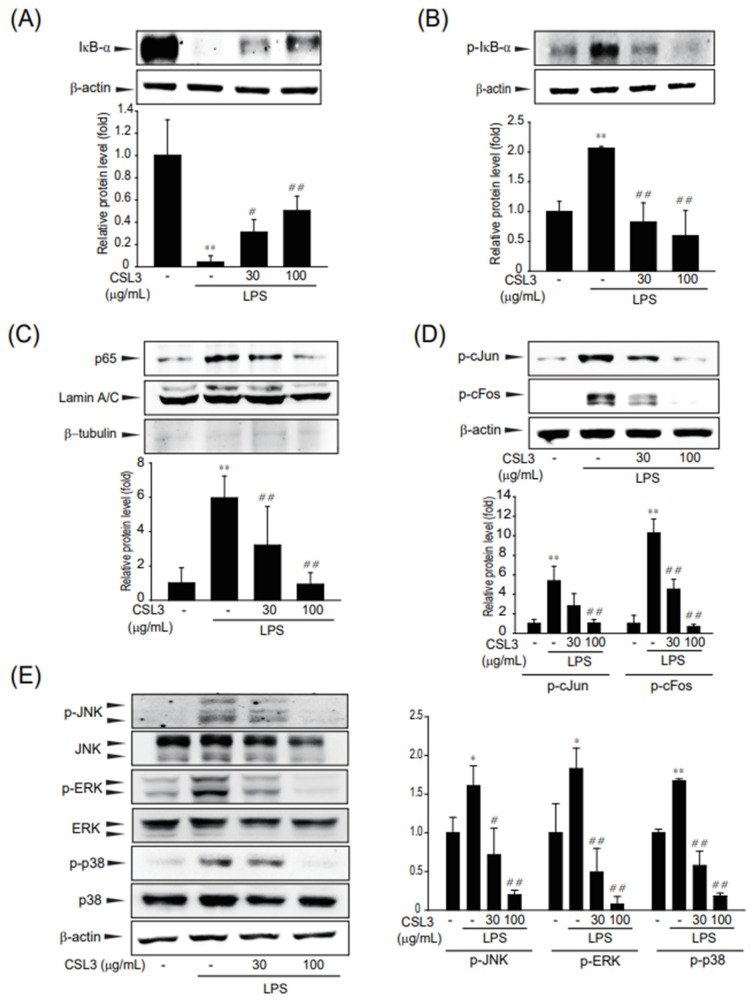
The enhibitory effect of CSL3 on NF-κB and AP-1 activation in LPS-activated Kupffer cells. (**A**,**B**) Immunoblotting for total (**A**) or phosphorylated IκB-α (**B**). The cells were pretreated with CSL3 for 1 h before LPS stimulation for 15 min. Total or phosphorylated IκB-α was immunoblotted in the cell lysate. (**C**) The expression level of p65 protein in ImKCs with nuclear fraction. Lamin A/C was a control for nuclear fraction and β-tubulin was a control for cytosolic fraction. ImKCs were pretreated with CSL3 for 1 h and then incubated with LPS for 30 min. (**D**) The expression of phosphorylated c-Jun and c-Fos protein in ImKCs. Cells were pretreated with CSL3 for 1 h before LPS stimulation for 2 h. c-Jun and c-Fos phosphorylation were immunoblotted with the cell lysate. (**E**) The expression level of MAPK phosphorylation in ImKCs. Cells were treated with LPS and/or CSL3 in ImKC cells for 1 h. MAPK proteins were visualized using immunoblotting. Data represent the mean ± S.E. of three replicates; ** *p* < 0.01, * *p* < 0.05, significant versus vehicle-treated control; ^##^
*p* < 0.01, ^#^
*p* < 0.05, significant versus LPS alone.

**Figure 4 ijms-24-11982-f004:**
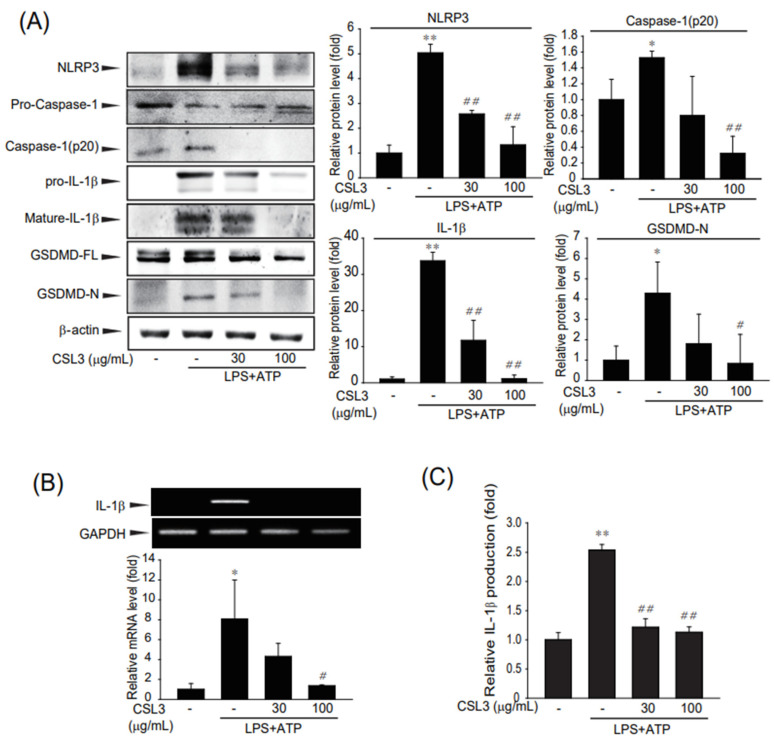
Attenuation of inflammasome-mediated pyroptosis by CSL3 in LPS and ATP-stimulated Kupffer cells. (**A**) Protein expression of inflammasome and pyroptosis markers were evaluated by immunoblotting. ImKC cells pretreated with CSL3 were incubated with LPS for 6 h and then treated with ATP for 0.5 h. (**B**) *Il-1β* mRNA levels were evaluated by RT-PCR using GAPDH as a control. Cells pretreated with CSL3 were incubated with LPS for 3 h and then treated with ATP for 0.5 h. (**C**) Enzyme-linked immunosorbent assay (ELISA). Cells pretreated with CSL3 were incubated with LPS for 3 h and then treated with ATP for 0.5 h. IL-1β levels were measured by an ELISA kit in the supernatant of ImKC cells. Data represent the mean ± S.E. of three replicates; ** *p* < 0.01, * *p* < 0.05, significant versus vehicle-treated control; ^##^
*p* < 0.01, ^#^
*p* < 0.05, significant versus LPS and ATP alone.

**Figure 5 ijms-24-11982-f005:**
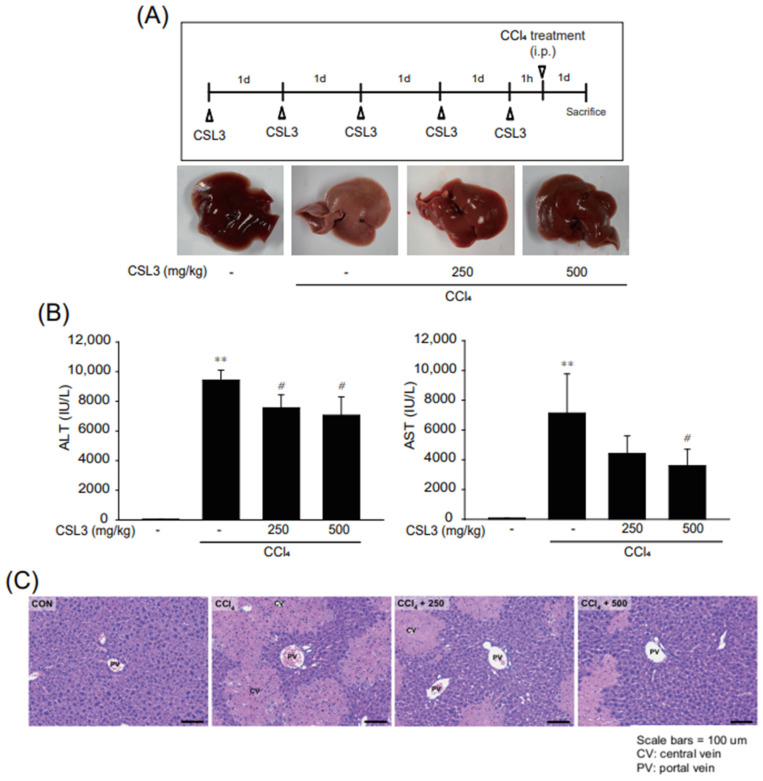
The hepatoprotective effect of CSL3 in CCl_4_-induced acute liver injury mice model. (**A**) Acute liver injury by intraperitoneal injection of CCl_4_ in male ICR mice. CSL3 was orally administered to the mice at a dose of 250 or 500 mg/kg/day for five consecutive days, followed by a subcutaneous injection of CCl_4_ (1 mL/kg, dissolved in olive oil (10%)) to induce acute liver injury; 24 h later, the mice were sacrificed. (**B**) Activities of serum ALT and AST were measured by using an automated blood chemistry analyzer. All values were expressed as mean ± S.E. of five mice serum (notable compared with vehicle control ** *p* < 0.01; notable compared with CCl_4_ alone, ^#^
*p* < 0.05). (**C**) Representative images of hematoxylin and eosin staining in the liver tissues. Scale bars indicate 100 μm.

**Figure 6 ijms-24-11982-f006:**
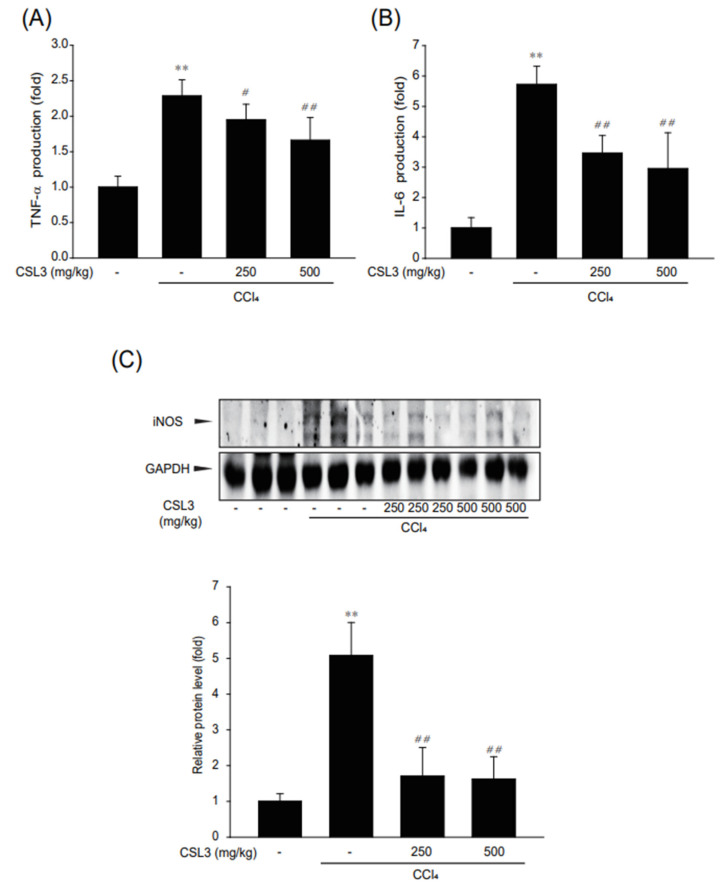
The inhibitory effect of CSL3 on inflammatory responses in CCl_4_-injected mice. (**A**,**B**) TNF-α and IL-6 release into serum was measured using ELISA. (**C**) The effect of CSL3 on CCl_4_-induced iNOS expression. All values were expressed as mean ± S.E. of three mice (notable compared with vehicle control, ** *p* < 0.01; notable compared with CCl_4_ alone, ^##^
*p* < 0.01, ^#^
*p* < 0.05).

**Figure 7 ijms-24-11982-f007:**
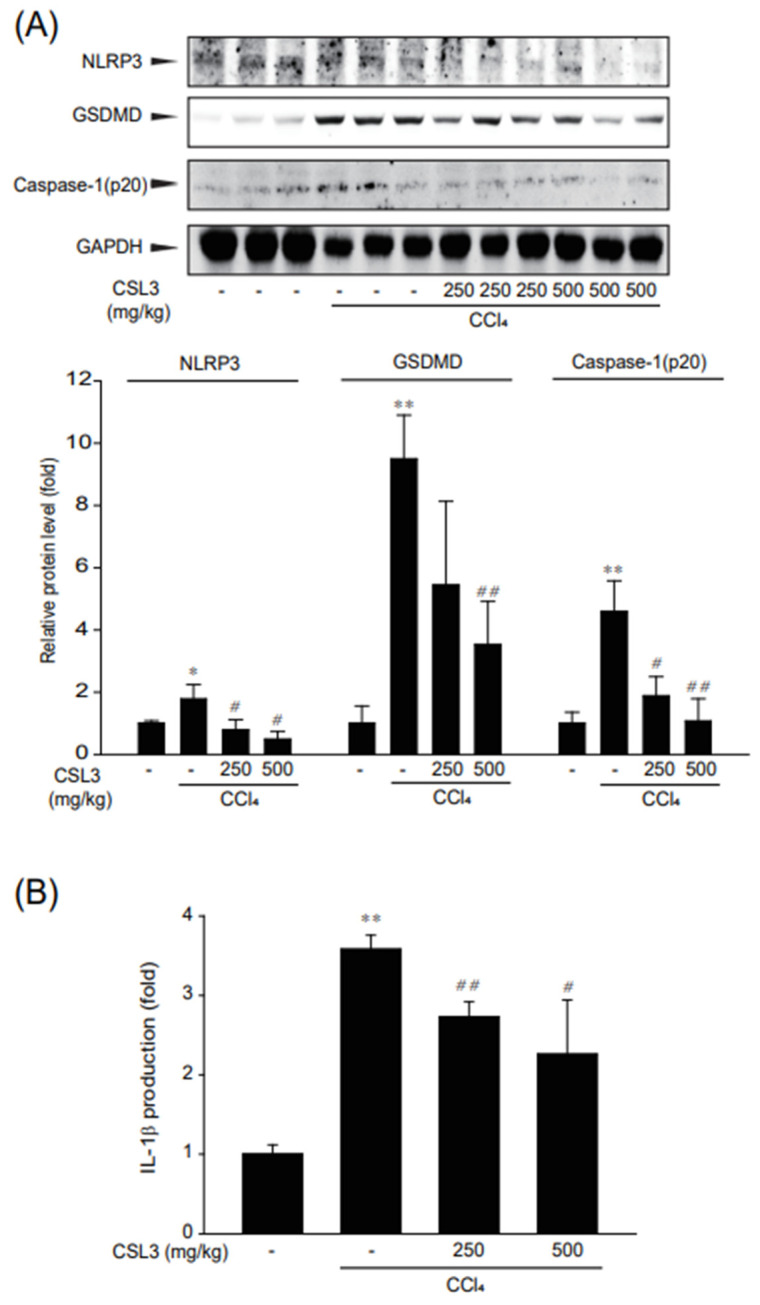
Suppression of CCl_4_-induced inflammasome and pyroptosis by CSL3. (**A**) Protein expression levels of inflammasome and pyroptosis markers were evaluated by immunoblotting using GAPDH as a control in liver tissue. (**B**) ELISA. IL-1β release into serum was established using an ELISA kit. All values were expressed as mean ± S.E. of three mice (notable compared with vehicle control, * *p* < 0.05, ** *p* < 0.01; notable compared with CCl_4_ alone, ^##^
*p* < 0.01, ^#^
*p* < 0.05).

**Figure 8 ijms-24-11982-f008:**
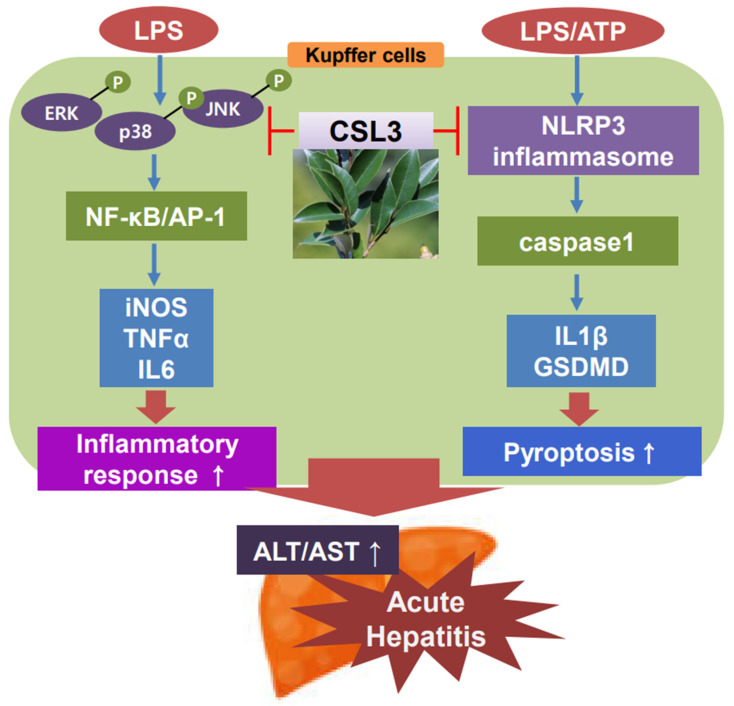
Schematic diagram illustrating the mechanism of CSL3 against acute hepatitis. (p: phosphorylation; ↑: increase).

## Data Availability

The data presented in this study are available in the article.

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
