# Peer review of "Castanopsis sieboldii Extract Alleviates Acute Liver Injury by Antagonizing Inflammasome-Mediated Pyroptosis"

_ijms, 2023, doi:10.3390/ijms241511982_

Round 1
Reviewer 1 Report
In the current manuscript, the authors investigated the effects of CSL3 on the lipopolysaccharide (LPS)-induced inflammatory responses and LPS and ATP-induced pyroptosis in macrophages. Their results suggested that CSL3 protects against acute liver injury by inhibiting inflammasome formation and pyroptosis. Although this study was well organized and clearly stated, there are still some limitations as described below.
1. The inhibition effect of CSL3 on iNOS expression and NO production was investigated in both LPS-activated ImKCs and Raw264.7 cells. However, other studies in vitro were only carried out using ImKCs cells. To enhance rigor, all the experiments in vitro should also be studied in Raw264.7 cells and supplemented in the manuscript.
2. Since the leaf extract of CS has high antioxidant activities, whether the antioxidant pathway is involved in the protection of CSL3 against acute liver injury should be studied
3. According to current results, it’s too early to draw the conclusion that the cytoprotective effect of CSL3 is due to the inhibition of inflammasome-mediated pyroptosis. Further experiments using appropriate inhibitors are required to support this conclusion.
4. How about the toxicity of CSL3 to other major organs under the dose of 250 or 500 mg/kg?
5. It would be better to add one schematic diagram to illustrate the mechanisms of CSL3.
Minors:
1. The experimental design of this article lacks some necessary references. For example, 1 mL/kg CCl4 was used to induce acute liver injury in mice and 0.1 μg/ml LPS was used to activate Kupffer cells. The related references should be supplemented to support the rational of the study.
2. In Fig 3D, there are no obvious differences between the control group and LPS-induced model group in the WB result of p-cJun, but the statistical differences were significant.
3. P or p for P value in the Fig legends should be unified.
4. The unit of L or l should also be united.
Minor editing of English language required.
Author Response
Reviewer #1:
1. The inhibition effect of CSL3 on iNOS expression and NO production was investigated in both LPS-activated ImKCs and Raw264.7 cells. However, other studies in vitro were only carried out using ImKCs cells. To enhance rigor, all the experiments in vitro should also be studied in Raw264.7 cells and supplemented in the manuscript.
Answer: We appreciate the reviewer’s helpful comments. It was well established that Kupffer cells (KCs), resident liver macrophages, play a major role in pathophysiology of acute liver injury. Thus, we have adopted ImKC cells, a murine immortalized Kupffer cell line to investigate the effect of CSL3 on acute liver injury and its molecular mechanism. We observed the comparable effect of CSL3 in Raw264.7 cells, most popular murine macrophage cell line, and then further experiments were conducted only in ImKC cells.
2. Since the leaf extract of CS has high antioxidant activities, whether the antioxidant pathway is involved in the protection of CSL3 against acute liver injury should be studied
Answer: Previously, we and our colleagues reported antioxidative effect of CLS3 as mentioned. In the present study, we have focused anti-inflammatory effect of CSL3 in liver resident macrophages. Moreover, we further investigate the effects of CSL3 and its major constituent in terms of antioxidative effect in hepatocytes against acute liver injury.
3. According to current results, it’s too early to draw the conclusion that the cytoprotective effect of CSL3 is due to the inhibition of inflammasome-mediated pyroptosis. Further experiments using appropriate inhibitors are required to support this conclusion.
Answer: CSL3 inhibits inflammatory response and inflammasome-mediated cell death, which might protect acute liver injury. Thus, further study using inhibitors to prove effect of CSL3 is not appropriate. However, further mechanism study was still required to determine CSL3 effect on inflammation-mediated acute liver injury. Limitation of current study was included in the revised MS.
4. How about the toxicity of CSL3 to other major organs under the dose of 250 or 500 mg/kg?
Answer: Food intake and body weight was monitored daily which were no difference between groups. Moreover, there were no signs of abnormality in other organs observed visually during necropsy.
5. It would be better to add one schematic diagram to illustrate the mechanisms of CSL3.
Answer: Schematic diagram to illustrate the mechanism of CSL3 against acute liver injury was included in the revised MS.
Minors:
1. The experimental design of this article lacks some necessary references. For example, 1 mL/kg CCl4 was used to induce acute liver injury in mice and 0.1 μg/ml LPS was used to activate Kupffer cells. The related references should be supplemented to support the rational of the study.
Answer: We have included the references of in vivo and in vitro experimental model used, as suggested.
2. In Fig 3D, there are no obvious differences between the control group and LPS-induced model group in the WB result of p-cJun, but the statistical differences were significant.
Answer: We have now replaced representative WB figure of p-cJun in the revised MS.
3. P or p for P value in the Fig legends should be unified.
Answer: We have corrected as recommended.
4. The unit of L or l should also be united.
Answer: We have edited as suggested.
Reviewer 2 Report
The topic is interesting and manuscript is well written with excellent figure included. However, there are certain points that could be addressed and improve the quality of this manuscript:
Delete the abbreviations in the abstract; which must be reported in the text only the first time they are cited and then make the list of abbreviations at the end of the manuscript
Did you use FBS medium in experiments where you treated the cells with CLS3?What about untreated cells and positive control? Please explain!
Line 260-263 CSL3 pretreatment inhibited the expression of NLRP3 an inflammasome sensor, and blocked he cleavage of caspase-1 and GSDMD. CSL3 also decreased production of pro-inflammatory cytokins, such as… how do you explain this? What about other studies? Compare please.
Line 269-271….Explain this results and compare with other studies.
Overall, I suggest in the discussion to mention other studies similar to your and to make a comparison!
Authors should include critical analysis with discussion of strengths and weaknesses of important studies that are cited.
Author Response
Reviewer #2:
The topic is interesting and manuscript is well written with excellent figure included. However, there are certain points that could be addressed and improve the quality of this manuscript:
1.Delete the abbreviations in the abstract; which must be reported in the text only the first time they are cited and then make the list of abbreviations at the end of the manuscript
Answer: We have corrected as suggested.
2.Did you use FBS medium in experiments where you treated the cells with CLS3? What about untreated cells and positive control? Please explain!
Answer: Cells were serum-starved for 12 h before treatment and control group was treated with vehicle correspondence.
3.Line 260-263 CSL3 pretreatment inhibited the expression of NLRP3 an inflammasome sensor, and blocked the cleavage of caspase-1 and GSDMD. CSL3 also decreased production of pro-inflammatory cytokins, such as… how do you explain this? What about other studies? Compare please.
Answer: There are several reports of some natural compounds with mechanism similar to CSL3. We have discussed in the revised MS.
4.Line 269-271….Explain this results and compare with other studies.
Overall, I suggest in the discussion to mention other studies similar to your and to make a comparison!
Answer: We have now included some literatures, which were similar to CSL3.
5.Authors should include critical analysis with discussion of strengths and weaknesses of important studies that are cited.
Answer: We have now included strengths and weaknesses of studies cited in the revised discussion.
Round 2
Reviewer 1 Report
The authors have addressed some points, but major issues remain unanswered and further experimental work is still lacking.
1. Since both ImKCs and Raw264.7 are macrophages, it is necessary to study and compare whether the activities of CSL3 are specific to the resident macrophages in liver tissue. Therefore, all the other experiments in Fig.2-Fig.4 should also be studied in Raw264.7 cells and supplemented in the manuscript.
2. The authors mentioned in the response that they further investigated the antioxidative effects of CSL3 in hepatocytes against acute liver injury. However, there is no data that could support this statement.
3. Genetic and biochemical methods are encouraged to be employed to determine CSL3 effect on inflammation-mediated acute liver injury. The authors stated in the response that the limitation of the current study was included in the revised MS. Which section?
Author Response
Reviewer #1:
The authors have addressed some points, but major issues remain unanswered and further experimental work is still lacking.
- Since both ImKCs and Raw264.7 are macrophages, it is necessary to study and compare whether the activities of CSL3 are specific to the resident macrophages in liver tissue. Therefore, all the other experiments in Fig.2-Fig.4 should also be studied in Raw264.7 cells and supplemented in the manuscript.
Answer: We appreciate the reviewer’s keen comments. We do not intend to compare whether CSL3 effect is specific to Kupffer cells. Therefore, anti-inflammatory efficacy by CSL3 were evaluated in both macrophages, and further experiments were conducted in Kupffer cells that reflect acute hepatitis. To avoid confusion raised by reviewer, we may remove result of RAW264.7 cell data if necessary. These were discussed in line 245-248 in the revised MS.
- The authors mentioned in the response that they further investigated the antioxidative effects of CSL3 in hepatocytes against acute liver injury. However, there is no data that could support this statement.
Answer: I’m sorry for the confusion. In the present study, we focused anti-inflammatory effect of CSL3 in macrophages. Further studies are thus still required to investigate the antioxidative efficacy and its concise molecular mechanism of CSL3 in in vitro and in vivo animal models for acute liver injury. Furthermore, we currently examine the antioxidative effects of CSL3 in hepatocytes apart from this project. These were discussed in line 298-300 in the revised MS.
- Genetic and biochemical methods are encouraged to be employed to determine CSL3 effect on inflammation-mediated acute liver injury. The authors stated in the response that the limitation of the current study was included in the revised MS. Which section?
Answer: Although we could observe hepatoprotective effect of CSL3 in Kupffer cells and CCl4-injected mice, there are some limitations in the current study. 1) We previously reported that CSL3 had the highest antioxidative capacity. We thus cannot exclude the possibility whether antioxidative efficacy of CSL3 might contribute to prevent acute hepatitis. 2) Albeit, CSL3 could succeed to inhibit inflammasome signaling pathway, we don’t have any direct clues whether CSL3 effect was due to inflammasome inhibition. Biochemical and genetic experimental models were still required to unveil concise molecular mechanism of CSL3. 3) It was reported that EGCG, major constituent of CSL3 inhibited inflammasome signaling pathway. However, further studies are necessary to define whether EGCG and other components might contribute to CSL3 effect against acute hepatitis. These are currently being conducted by this research team as a separate project. These were discussed in line 297-307 in the revised MS.
Round 3
Reviewer 1 Report
The authors have addressed my comments.